# Simple random search of static linear policies is competitive for reinforcement learning

**Horia Mania**
hmania@berkeley.edu

**Aurelia Guy**
lia@berkeley.edu

**Benjamin Recht**
brecht@berkeley.edu

Department of Electrical Engineering and Computer Science
University of California, Berkeley

## Abstract

Model-free reinforcement learning aims to offer off-the-shelf solutions for controlling dynamical systems without requiring models of the system dynamics. We introduce a model-free random search algorithm for training static, linear policies for continuous control problems. Common evaluation methodology shows that our method matches state-of-the-art sample efficiency on the benchmark MuJoCo locomotion tasks. Nonetheless, more rigorous evaluation reveals that the assessment of performance on these benchmarks is optimistic. We evaluate the performance of our method over hundreds of random seeds and many different hyperparameter configurations for each benchmark task. This extensive evaluation is possible because of the small computational footprint of our method. Our simulations highlight a high variability in performance in these benchmark tasks, indicating that commonly used estimations of sample efficiency do not adequately evaluate the performance of RL algorithms. Our results stress the need for new baselines, benchmarks and evaluation methodology for RL algorithms.

## 1   Introduction

Model-free reinforcement learning (RL) aims to offer off-the-shelf solutions for controlling dynamical systems without requiring models of the system dynamics. Such methods have successfully produced RL agents that surpass human players in video games and games such as Go [16, 28]. Although these results are impressive, model-free methods have not yet been successfully deployed to control physical systems, outside of research demos. There are several factors prohibiting the adoption of model-free RL methods for controlling physical systems: the methods require too much data to achieve reasonable performance, the ever-increasing assortment of RL methods makes it difficult to choose what is the best method for a specific task, and many candidate algorithms are difficult to implement and deploy [11].

Unfortunately, the current trend in RL research has put these impediments at odds with each other. In the quest to find methods that are *sample efficient* (i.e. methods that need little data) the general trend has been to develop increasingly complicated methods. This increasing complexity has led to a reproducibility crisis. Recent studies demonstrate that many RL methods are not robust to changes in hyperparameters, random seeds, or even different implementations of the same algorithm [11, 12]. Algorithms with such fragilities cannot be integrated into mission critical control systems without significant simplification and robustification.

Furthermore, it is common practice to evaluate and compare new RL methods by applying them to video games or simulated continuous control problems and measure their performance over a small number of independent trials (i.e., fewer than ten random seeds) [8–10, 14, 17, 19, 21–27, 31, 32]. The most popular continuous control benchmarks are the MuJoCo locomotion tasks [3, 29], with

the Humanoid model being considered "one of the most challenging continuous control problems solvable by state-of-the-art RL techniques [23]." In principle, one can use video games and simulated control problems for beta testing new ideas, but simple baselines should be established and thoroughly evaluated before moving towards more complex solutions.

To this end, we aim to determine *the simplest* model-free RL method that can solve standard benchmarks. Recently, two different directions have been proposed for simplifying RL. Salimans et al. [23] introduced a derivative-free policy optimization method, called Evolution Strategies. The authors showed that, for several RL tasks, their method can easily be parallelized to train policies faster than other methods. While the method of Salimans et al. [23] is simpler than previously proposed methods, it employs several complicated algorithmic elements, which we discuss at the end of Section 3. As a second simplification to model-free RL, Rajeswaran et al. [22] have shown that linear policies can be trained via natural policy gradients to obtain competitive performance on the MuJoCo locomotion tasks, showing that complicated neural network policies are not needed to solve these continuous control problems. In this work, we combine ideas from the work of Salimans et al. [23] and Rajeswaran et al. [22] to obtain the simplest model-free RL method yet, a derivative-free optimization algorithm for training static, linear policies. We demonstrate that a simple random search method can match or exceed state-of-the-art sample efficiency on the MuJoCo locomotion tasks, included in the OpenAI Gym.

Henderson et al. [11] and Islam et al. [12] pointed out that standard evaluation methodology does not accurately capture the performance of RL methods by showing that existing RL algorithms exhibit high sensitivity to both the choice of random seed and the choice of hyperparameters. We show similar limitations of common evaluation methodology through a different lens. We exhibit a simple derivative free optimization algorithm which matches or surpasses the performance of more complex methods when using the same evaluation methodology. However, a more thorough evaluation of ARS reveals worse performance. Moreover, our method uses static linear policies and a simple local exploration scheme, which might be limiting for more difficult RL tasks. Therefore, better evaluation schemes are needed for determining the benefits of more complex RL methods. Our contributions are as follows:

- In Section 3, for applications to continuous control, we augment a basic random search method with three simple features. First, we scale each update step by the standard deviation of the rewards collected for computing that update step. Second, we normalize the system's states by online estimates of their mean and standard deviation. Third, we discard from the computation of the update steps the directions that yield the least improvement of the reward. We refer to this method as *Augmented Random Search* (ARS).

- In Section 4, we evaluate the performance of ARS on the benchmark MuJoCo locomotion tasks, included in the OpenAI Gym. Our method learns static, linear policies that achieve high rewards on all MuJoCo tasks. No neural networks are used, and yet state-of-the-art average rewards are achieved. For example, for Humanoid-v1 ARS finds linear policies which achieve average rewards of over 11500, the highest value reported in the literature. To put ARS on equal footing with competing methods, we evaluate its sample complexity over three random seeds and compare it to results reported in the literature [9, 22, 23, 26]. ARS matches or exceeds state-of-the-art sample efficiency on the locomotion tasks when using standard evaluation methodology.

- For a more thorough evaluation, we measured the performance of ARS over a hundred random seeds and also evaluated its sensitivity to hyperparameter choices. Though ARS successfully trains policies for the MuJoCo tasks a large fraction of the time when hyperparameters and random seeds are varied, ARS exhibits large variance. We measure the frequency with which ARS finds policies that yield suboptimal locomotion gaits.

## 2   Problem setup

Problems in reinforcement learning require finding policies for controlling dynamical systems that maximize an average reward. Such problems can be abstractly formulated as

$$\max_{\theta \in \mathbb{R}^d} \mathbb{E}_\xi \left[ r(\pi_\theta, \xi) \right] , \tag{1}$$

where $\theta$ parametrizes a policy $\pi_\theta : \mathbb{R}^n \to \mathbb{R}^p$. The random variable $\xi$ encodes the randomness of the environment, i.e., random initial states and stochastic transitions. The value $r(\pi_\theta, \xi)$ is the reward achieved by the policy $\pi_\theta$ on one trajectory generated from the system. In general one could use stochastic policies $\pi_\theta$, but our proposed method uses deterministic policies.

**Basic random search.**   Note that the problem formulation (1) aims to optimize reward by directly optimizing over the policy parameters $\theta$. We consider methods which explore in the parameter space rather than the action space. This choice renders RL training equivalent to derivative-free optimization with noisy function evaluations. One of the simplest and oldest optimization methods for derivative-free optimization is *random search* [15].

A primitive form of random search, which we call *basic random search* (BRS), simply computes a finite difference approximation along the random direction and then takes a step along this direction without using a line search. Our method ARS, described in Section 3, is based on this simple strategy. For updating the parameters $\theta$ of a policy $\pi_\theta$, BRS and ARS exploit update directions of the form:

$$\frac{r(\pi_{\theta+\nu\delta}, \xi_1) - r(\pi_{\theta-\nu\delta}, \xi_2)}{\nu}, \tag{2}$$

for two i.i.d. random variables $\xi_1$ and $\xi_2$, $\nu$ a positive real number, and $\delta$ a zero mean Gaussian vector. It is known that such an update increment is an unbiased estimator of the gradient with respect to $\theta$ of $\mathbb{E}_\delta \mathbb{E}_\xi \left[ r(\pi_{\theta+\nu\delta}, \xi) \right]$, a smoothed version of the objective (1) which is close to the original objective when $\nu$ is small [20]. When the function evaluations are noisy, minibatches can be used to reduce the variance in this gradient estimate. Evolution Strategies is a version of this algorithm with several complicated algorithmic enhancements [23]. Another version of this algorithm is called Bandit Gradient Descent by Flaxman et al. [6]. The convergence of random search methods for derivative free optimization has been understood for several types of convex optimization [1, 2, 13, 20]. Jamieson et al. [13] offer an information theoretic lower bound for derivative free convex optimization and show that a coordinate based random search method achieves the lower bound with nearly optimal dependence on the dimension.

The rewards $r(\pi_{\theta+\nu\delta}, \xi_1)$ and $r(\pi_{\theta-\nu\delta}, \xi_2)$ in Eq. (2) are obtained by collecting two trajectories from the dynamical system of interest, according to the policies $\pi_{\theta+\nu\delta}$ and $\pi_{\theta-\nu\delta}$, respectively. The random variables $\xi_1$, $\xi_2$, and $\delta$ are mutually independent, and independent from previous trajectories. One trajectory is called an *episode* or a *rollout*. The goal of RL algorithms is to approximately solve problem (1) by using as few rollouts from the dynamical system as possible.

## 3   Our proposed algorithm

We now introduce the Augmented Random Search (ARS) method, which relies on three augmentations of BRS that build on successful heuristics employed in deep reinforcement learning. Throughout the rest of the paper we use $M$ to denote the parameters of policies because our method uses linear policies, and hence $M$ is a $p \times n$ matrix. The different versions of ARS are detailed in Algorithm 1.

The first version, ARS **V1**, is obtained from BRS by scaling the update steps by the standard deviation $\sigma_R$ of the rewards collected at each iteration; see Line 7 of Algorithm 1. As shown in Section 4, ARS **V1** can train linear policies, which achieve the reward thresholds previously proposed in the literature, for five MuJoCo benchmarks. However, ARS **V1** requires a larger number of episodes, and it cannot train policies for the Humanoid-v1 task. To address these issues in Algorithm 1 we also propose ARS **V2**. This version of ARS trains policies which are linear maps of states normalized by a mean and standard deviation computed online. Finally, to further enhance the performance of ARS, we introduce a third algorithmic enhancement, shown in Algorithm 1 as ARS **V1-t** and ARS **V2-t**. These versions of ARS can drop perturbation directions that yield the least improvement of the reward. Now, we motivate and offer intuition for each of these algorithmic elements.

**Scaling by the standard deviation $\sigma_R$.**   As the training of policies progresses, random search in the parameter space of policies can lead to large variations in the rewards observed across iterations. As a result, it is difficult to choose a fixed step-size $\alpha$ which does not allow harmful variations in the size of the update steps. Salimans et al. [23] address this issue by transforming the rewards into rankings and then using the adaptive optimization algorithm Adam for computing the update step. Both of

---

**Algorithm 1** Augmented Random Search (ARS): four versions **V1**, **V1-t**, **V2** and **V2-t**

---

1: **Hyperparameters:** step-size $\alpha$, number of directions sampled per iteration $N$, standard deviation of the exploration noise $\nu$, number of top-performing directions to use $b$ ($b < N$ is allowed only for **V1-t** and **V2-t**)

2: **Initialize:** $M_0 = \mathbf{0} \in \mathbb{R}^{p \times n}$, $\mu_0 = \mathbf{0} \in \mathbb{R}^n$, and $\Sigma_0 = \mathbf{I}_n \in \mathbb{R}^{n \times n}$, $j = 0$.

3: **while** ending condition not satisfied **do**

4:     Sample $\delta_1, \delta_2, \ldots, \delta_N$ in $\mathbb{R}^{p \times n}$ with i.i.d. standard normal entries.

5:     Collect $2N$ rollouts of horizon $H$ and their corresponding rewards using the $2N$ policies

$$\textbf{V1:} \begin{cases} \pi_{j,k,+}(x) = (M_j + \nu\delta_k)x \\ \pi_{j,k,-}(x) = (M_j - \nu\delta_k)x \end{cases}$$

$$\textbf{V2:} \begin{cases} \pi_{j,k,+}(x) = (M_j + \nu\delta_k)\operatorname{diag}(\Sigma_j)^{-1/2}(x - \mu_j) \\ \pi_{j,k,-}(x) = (M_j - \nu\delta_k)\operatorname{diag}(\Sigma_j)^{-1/2}(x - \mu_j) \end{cases}$$

    for $k \in \{1, 2, \ldots, N\}$.

6:     **V1-t, V2-t:** Sort the directions $\delta_k$ by $\max\{r(\pi_{j,k,+}), r(\pi_{j,k,-})\}$, denote by $\delta_{(k)}$ the $k$-th largest direction, and by $\pi_{j,(k),+}$ and $\pi_{j,(k),-}$ the corresponding policies.

7:     Make the update step:

$$M_{j+1} = M_j + \frac{\alpha}{b\sigma_R} \sum_{k=1}^{b} \left[ r(\pi_{j,(k),+}) - r(\pi_{j,(k),-}) \right] \delta_{(k)},$$

    where $\sigma_R$ is the standard deviation of the $2b$ rewards used in the update step.

8:     **V2:** Set $\mu_{j+1}, \Sigma_{j+1}$ to be the mean and covariance of the $2NH(j+1)$ states encountered from the start of training.[1]

9:     $j \leftarrow j + 1$

10: **end while**

.

---

these techniques change the direction of the updates, obfuscating the behavior of the algorithm and making it difficult to ascertain the objective Evolution Strategies is actually optimizing. Instead, to address the large variations of the differences $r(\pi_{M+\nu\delta}) - r(\pi_{M-\nu\delta})$, we scale the update steps by the standard deviation $\sigma_R$ of the $2N$ rewards collected at each iteration (see Line 7 of Algorithm 1).

While training a policy for Humanoid-v1, we observed that the standard deviations $\sigma_R$ have an increasing trend; see Figure 2 in Appendix A.2. This behavior occurs because perturbations of the policy weights at high rewards can cause Humanoid-v1 to fall early, yielding large variations in the rewards collected. Without scaling the update steps by $\sigma_R$, eventually random search would take update steps which are a thousand times larger than in the beginning of training. Therefore, $\sigma_R$ adapts the step sizes according to the local sensitivity of the rewards to perturbations of the policy parameters. The same training performance could probably be obtained by tuning a step size schedule. However, one of our goals was to minimize the amount of tuning required.

**Normalization of the states.** The normalization of states used by ARS **V2** is akin to data whitening for regression tasks. Intuitively, it ensures that policies put equal weight on the different components of the states. To see why this might help, suppose that a state coordinate only takes values in the range $[90, 100]$ while another state component takes values in the range $[-1, 1]$. Then, small changes in the control gain with respect to the first state coordinate would lead to larger changes in the actions than the same sized changes with respect to the second state component. Hence, state normalization allows different state components to have equal influence during training.

Previous work has also implemented such state normalization for fitting a neural network model for several MuJoCo environments [19]. A similar normalization is used by ES as part of the virtual batch

normalization of the neural network policies [23]. In the case of ARS, the state normalization can be seen as a form of non-isotropic exploration in the parameter space of linear policies.

The main empirical motivation for ARS **V2** comes from the Humanoid-v1 task. We were not able to train a linear policy for this task without the normalization of the states described in Algorithm 1. Moreover, ARS **V2** performs better than ARS **V1** on other MuJoCo tasks as well, as shown in Section 4. However, the usefulness of state normalization is likely to be problem specific.

**Using top performing directions.** To further improve the performance of ARS on the MuJoCo locomotion tasks, we propose ARS **V1-t** and **V2-t**. In the update steps used by ARS **V1** and **V2** each perturbation direction $\delta_k$ is weighted by the difference of the rewards $r(\pi_{j,k,+})$ and $r(\pi_{j,k,-})$. If $r(\pi_{j,k,+}) > r(\pi_{j,k,-})$, ARS pushes the policy weights $M_j$ in the direction of $\delta_k$. If $r(\pi_{j,k,+}) < r(\pi_{j,k,-})$, ARS pushes the policy weights $M_j$ in the direction of $-\delta_k$. However, since $r(\pi_{j,k,+})$ and $r(\pi_{j,k,-})$ are noisy evaluations of the performance of the policies parametrized by $M_j + \nu\delta_k$ and $M_j - \nu\delta_k$, ARS **V1** and **V2** might push the weights $M_j$ in the direction $\delta_k$ even when $-\delta_k$ is better, or vice versa. Moreover, there can be perturbation directions $\delta_k$ such that updating the policy weights $M_j$ in either the direction $\delta_k$ or $-\delta_k$ would lead to sub-optimal performance. To address these issues, ARS **V1-t** and **V2-t** order decreasingly the perturbation directions $\delta_k$, according to $\max\{r(\pi_{j,k,+}), r(\pi_{j,k,-})\}$, and then use only the top $b$ directions for updating the policy weights; see Line 7 of Algorithm 1.

This algorithmic enhancement intuitively improves the performance of ARS because it ensures that the update steps are an average over directions that obtained high rewards. However, without theoretical investigation we cannot be certain of the effect of using this algorithmic enhancement, i.e., choosing $b < N$. When $b = N$ versions **V1-t** and **V2-t** are equivalent to **V1** and **V2**. Therefore, it is certain that after tuning ARS **V1-t** and **V2-t**, they will not perform any worse than ARS **V1** and **V2**.

**Comparison to Salimans et al. [23].** ARS simplifies Evolution Strategies in several ways. First, ES feeds the gradient estimate into the Adam algorithm. Second, instead of using the actual reward values $r(\theta \pm \sigma\epsilon_i)$, ES transforms the rewards into rankings and uses the ranks to compute update steps. The rankings are used to make training more robust. Instead, our method scales the update steps by the standard deviation of the rewards. Third, ES bins the action space of the Swimmer-v1 and Hopper-v1 to encourage exploration. Our method surpasses ES without such binning. Fourth, ES relies on policies parametrized by neural networks with virtual batch normalization, while we show that ARS achieves state-of-the-art performance with linear policies.

## 4 Empirical results on the MuJoCo locomotion tasks

**Implementation details.** We implemented a parallel version of Algorithm 1 using the Python library Ray [18]. To avoid the computational bottleneck of communicating perturbations $\delta$, we created a shared noise table which stores independent standard normal entries. Then, instead of communicating perturbations $\delta$, the workers communicate indices in the shared noise table. This approach has been used in the implementation of Evolution Strategies by Moritz et al. [18] and is similar to the approach proposed by Salimans et al. [23]. Our code sets the random seeds for the random generators of all the workers and for all copies of the OpenAI Gym environments held by the workers. All these random seeds are distinct and are a function of a single integer to which we refer as *the random seed*. Furthermore, we made sure that the states and rewards produced during the evaluation rollouts were not used in any form during training.

We evaluate the performance of ARS on the MuJoCo locomotion tasks included in the OpenAI Gym-v0.9.3 [3, 29]. The OpenAI Gym provides benchmark reward functions for the different MuJoCo locomotion tasks. We used these default reward functions for evaluating the performance of the linear policies trained with ARS. The reported rewards obtained by a policy were averaged over 100 independent rollouts. For the Hopper-v1, Walker2d-v1, Ant-v1, and Humanoid-v1 tasks the default reward functions include a survival bonus, which rewards RL agents with a constant reward at each timestep, as long as a termination condition (i.e., falling over) has not been reached. During training, we removed these survival bonuses, a choice we motivate in Appendix A.1. We also defer to Appendix A.3 the sensitivity analysis of ARS to the choice of hyperparameters.

**Three random seeds evaluation:** We compare the different versions of ARS to the following methods: Trust Region Policy Optimization (TRPO), Deep Deterministic Policy Gradient (DDPG), Natural Gradients (NG), Evolution Strategies (ES), Proximal Policy Optimization (PPO), Soft Actor Critic (SAC), Soft Q-Learning (SQL), A2C, and the Cross Entropy Method (CEM). For the performance of these methods we used values reported by Rajeswaran et al. [22], Salimans et al. [23], Schulman et al. [26], and Haarnoja et al. [9]. In light of well-documented reproducibility issues of reinforcement learning methods [11, 12], reporting the values listed in papers rather than rerunning these algorithms casts prior work in the most favorable light possible.

Rajeswaran et al. [22] and Schulman et al. [26] evaluated the performance of RL algorithms on three random seeds, while Salimans et al. [23] and Haarnoja et al. [9] used six and five random seeds respectively. To put all methods on equal footing, for the evaluation of ARS, we sampled three random seeds uniformly from the interval $[0, 1000)$ and fixed them. For each of the six OpenAI Gym MuJoCo locomotion tasks we chose a grid of hyperparameters[2], shown in Appendix A.6, and for each set of hyperparameters we ran ARS **V1**, **V2**, **V1-t**, and **V2-t** three times, once for each of the three fixed random seeds.

Table 1 shows the average number of episodes required by ARS, NG, and TRPO to reach a prescribed reward threshold, using the values reported by Rajeswaran et al. [22] for NG and TRPO. For each version of ARS and each MuJoCo task we chose the hyperparameters which minimize the average number of episodes required to reach the reward threshold. The corresponding training curves of ARS are shown in Figure 3 of Appendix A.2. For all MuJoCo tasks, except Humanoid-v1, we used the same reward thresholds as Rajeswaran et al. [22]. Our choice to increase the reward threshold for Humanoid-v1 is motivated by the presence of the survival bonuses, as discussed in Appendix A.1.

| Task | Threshold | Average # episodes to reach reward threshold | | | | NG-lin | NG-rbf | TRPO-nn |
|---|---|---|---|---|---|---|---|---|
| | | ARS | | | | | | |
| | | V1 | V1-t | V2 | V2-t | | | |
| Swimmer-v1 | 325 | 100 | 100 | 427 | 427 | 1450 | 1550 | N/A[3] |
| Hopper-v1 | 3120 | 89493 | 51840 | 3013 | 1973 | 13920 | 8640 | 10000 |
| HalfCheetah-v1 | 3430 | 10240 | 8106 | 2720 | 1707 | 11250 | 6000 | 4250 |
| Walker2d-v1 | 4390 | 392000 | 166133 | 89600 | 24000 | 36840 | 25680 | 14250 |
| Ant-v1 | 3580 | 101066 | 58133 | 60533 | 20800 | 39240 | 30000 | 73500 |
| Humanoid-v1 | 6000 | N/A | N/A | 142600 | 142600 | ≈130000 | ≈130000 | UNK[4] |

**Table 1:** A comparison of ARS, NG, and TRPO on the MuJoCo locomotion tasks. For each task we show the average number of episodes required to achieve a prescribed reward threshold, averaged over three random seeds. We estimated the number of episodes required by NG to reach a reward of 6000 for Humanoid-v1 based on the learning curves presented by Rajeswaran et al. [22].

Table 1 shows that ARS **V1** can train policies for all tasks except Humanoid-v1, which is successfully solved by ARS **V2**. Secondly, we note that ARS **V2** reaches the prescribed thresholds for Swimmer-v1, Hopper-v1, and HalfCheetah-v1 faster than NG or TRPO, and matches the performance of NG on the Humanoid-v1. On Walker2d-v1 and Ant-v1, ARS **V2** is outperformed by NG. Nonetheless, ARS **V2-t** surpasses the performance of NG on these two tasks. Although TRPO hits the reward threshold for Walker2d-v1 faster than ARS, our method either matches or surpasses TRPO in the metrics reported by Haarnoja et al. [9] and Schulman et al. [26].

Precise comparisons to more RL methods are provided in Appendix A.2. Here we offer a summary. Salimans et al. [23] reported the average number of episodes required by ES to reach a prescribed reward threshold, on four of the locomotion tasks. ARS surpassed ES on all of those tasks. Haarnoja et al. [9] reported the maximum reward achieved by SAC, DDPG, SQL, and TRPO after a prescribed number of timesteps, on four of the locomotion tasks. With the exception of SAC on HalfCheetah-v1 and Ant-v1, ARS outperformed competing methods. Schulman et al. [26] reported the maximum reward achieved by PPO, A2C, CEM, and TRPO after a prescribed number of timesteps, on four of

the locomotion tasks. With the exception of PPO on Walker2d-v1, ARS matched or surpassed the performance of competing methods.

**A hundred seeds evaluation:** For a more thorough evaluation of ARS, we sampled 100 distinct random seeds uniformly at random from the interval $[0, 10000)$. Then, using the hyperparameters selected for Table 1, we ran ARS for each of the six MuJoCo locomotion tasks and the 100 random seeds. The results are shown in Figure 1. Such a thorough evaluation was feasible because ARS has a small computational footprint. As discussed in Appendix A.3, ARS is at least 15 times more computationally efficient on the MuJoCo benchmarks than competing methods.

Figure 1 shows that 70% of the time ARS trains policies for all the MuJoCo locomotion tasks, with the exception of Walker2d-v1 for which it succeeds only 20% of the time. Moreover, ARS succeeds at training policies a large fraction of the time while using a competitive number of episodes.

**Average reward evaluated over 100 random seeds, shown by percentile**

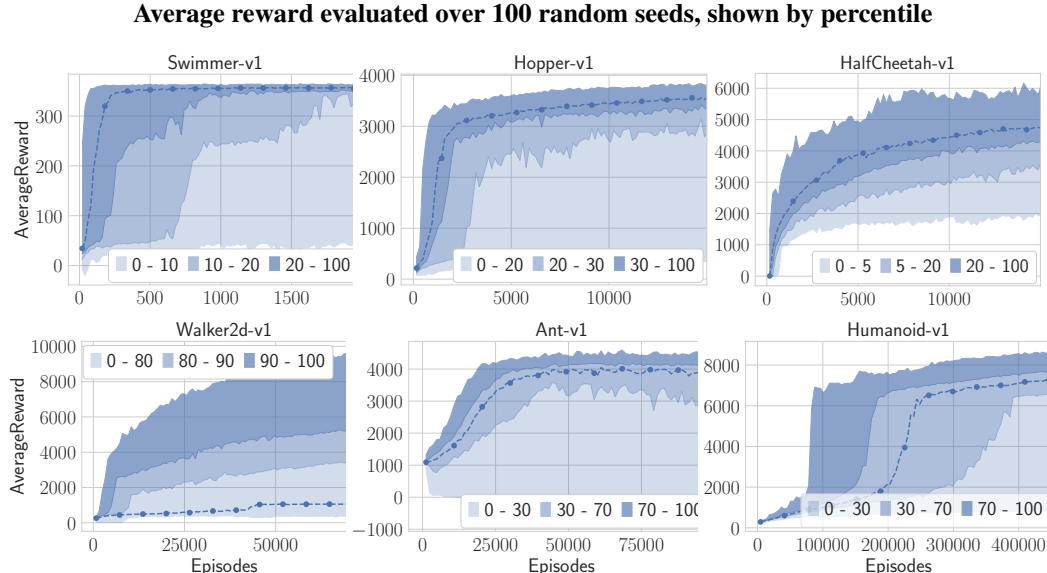

**Figure 1:** An evaluation of ARS over 100 random seeds on the MuJoCo locomotion tasks. The dotted lines represent median rewards and the shaded regions represent percentiles. For Swimmer-v1 we used ARS **V1**. For Hopper-v1, Walker2d-v1, and Ant-v1 we used ARS **V2-t**. For HalfCheetah-v1 and Humanoid-v1 we used ARS **V2**.

There are two types of random seeds represented in Figure 1 that cause ARS to not reach high rewards. There are random seeds on which ARS eventually finds high reward policies when sufficiently many iterations of ARS are performed, and there are random seeds which lead ARS to discover locally optimal behaviors. For the Humanoid model, ARS found numerous distinct gaits, including ones during which the Humanoid hops only on one leg, walks backwards, or moves in a swirling motion. Such gaits were found by ARS on the random seeds which cause slower training. While multiple gaits for Humanoid models have been previously observed [10], our evaluation better emphasizes their prevalence. The presence of local optima is inherent to non-convex optimization, and our results show that RL algorithms should be evaluated on many random seeds for determining the frequency with which local optima are found. Finally, we remark that ARS is the least sensitive to the choice of random seed used when applied to HalfCheetah-v1, a task which is often used for the evaluation of sensitivity of algorithms to the choice of random seeds.

**Linear policies are sufficiently expressive for MuJoCo:** We discussed how linear policies can produce diverse gaits for the MuJoCo models, showing that they are sufficiently expressive to capture diverse behaviors. Table 2 shows that linear policies can also achieve high rewards on all the MuJoCo locomotion tasks. In particular, for Humanoid-v1 and Walker2d-v1, ARS found policies that achieve significantly higher rewards than any other results we encountered in the literature. These results show that linear policies are perfectly adequate for the MuJoCo locomotion tasks, reducing the need for more expressive and more computationally expensive policies.

**Maximum reward achieved**

| Task | ARS | Task | ARS | Task | ARS |
|------|-----|------|-----|------|-----|
| Swimmer-v1 | 365 | HalfCheetah-v1 | 6722 | Ant | 5146 |
| Hopper-v1 | 3909 | Walker | 11389 | Humanoid | 11600 |

**Table 2:** Maximum average reward achieved by ARS, where we took the maximum over all sets of hyperparameters considered and the three fixed random seeds.

## 5  Discussion

With a few algorithmic augmentations, basic random search of static, linear policies achieves state-of-the-art sample efficiency on the MuJoCo locomotion tasks. Surprisingly, no special nonlinear controllers are needed to match the performance recorded in the RL literature. Moreover, since our algorithm and policies are simple, we were able to perform extensive sensitivity analysis. This analysis brings us to an uncomfortable conclusion that the current evaluation methods adopted in the deep RL community are insufficient to evaluate whether proposed methods are actually solving the studied problems.

The choice of benchmark tasks and the small number of random seeds do not represent the only issues of current evaluation methodology. Though many RL researchers are concerned about minimizing sample complexity, *it does not make sense to optimize the running time of an algorithm on a single problem instance.* The running time of an algorithm is only a meaningful notion if either (a) evaluated on a family of problem instances, or (b) when clearly restricting the class of algorithms.

Common RL practice, however, does not follow either (a) or (b). Instead, researchers run an algorithm $\mathcal{A}$ on a task $\mathcal{T}$ with a given hyperparameter configuration, and plot a "learning curve" showing the algorithm reaches a target reward after collecting $X$ samples. Then the "sample complexity" of the method is reported as the number of samples required to reach a target reward threshold, with the given hyperparameter configuration. However, any number of hyperparameter configurations can be tried. Any number of algorithmic enhancements can be added or discarded and then tested in simulation. For a fair measurement of sample complexity, should we not count the number of rollouts used for all tested hyperparameters?

Through optimal hyperparameter tuning one can artificially improve the perceived sample efficiency of a method. Indeed, this is what we see in our work. By adding a third algorithmic enhancement to basic random search (i.e., enhancing ARS **V2** to **V2-t**), we are able to improve the sample efficiency of an already highly performing method. Considering that most of the prior work in RL uses algorithms with far more tunable parameters and neural nets whose architectures themselves are hyperparameters, the significance of the reported sample complexities for those methods is not clear. This issue is important because a meaningful sample complexity of an algorithm should inform us on the number of samples required to solve a new, previously unseen task.

In light of these issues and of our empirical results, we make several suggestions for future work:

- Simple baselines should be established before moving forward to more complex benchmarks and methods. We propose the Linear Quadratic Regulator as a reasonable testbed for RL algorithms. LQR is well-understood when the model is known, problem instances can be easily generated with a variety of different levels of difficulty, and little overhead is required for replication; see Appendix A.4 for more details.

- When games and physics simulators are used for evaluation, separate problem instances should be used for tuning and evaluating RL methods. Moreover, large numbers of random seeds should be used for statistically significant evaluations.

- Rather than trying to develop general purpose algorithms, it might be better to focus on specific problems of interest and find targeted solutions.

- More emphasis should be put on the development of model-based methods. For many problems, such methods have been observed to require fewer samples than model-free methods. Moreover, the physics of the systems should inform the parametric classes of models used for different problems. Model-based methods incur many computational challenges themselves, and it is quite possible that tools from deep RL, such as improved

tree search, can provide new paths forward for tasks that require the navigation of complex and uncertain environments.

## Acknowledgments

We thank Orianna DeMasi, Moritz Hardt, Eric Jonas, Robert Nishihara, Rebecca Roelofs, Esther Rolf, Vaishaal Shankar, Ludwig Schmidt, Nilesh Tripuraneni, Stephen Tu for many helpful comments and suggestions. HM thanks Robert Nishihara and Vaishaal Shankar for sharing their expertise in parallel computing. As part of the RISE lab, HM is generally supported in part by NSF CISE Expeditions Award CCF-1730628, DHS Award HSHQDC-16-3-00083, and gifts from Alibaba, Amazon Web Services, Ant Financial, CapitalOne, Ericsson, GE, Google, Huawei, Intel, IBM, Microsoft, Scotiabank, Splunk and VMware. BR is generously supported in part by NSF award CCF-1359814, ONR awards N00014-14-1-0024 and N00014-17-1-2191, the DARPA Fundamental Limits of Learning (Fun LoL) Program, and an Amazon AWS AI Research Award.

## Footnotes

[1] Of course, we implement this in an efficient way that does not require the storage of all the states. Also, we only keep track of the diagonal of $\Sigma_{j+1}$. Finally, to ensure that the ratio $0/0$ is treated as 0, if a diagonal entry of $\Sigma_j$ is smaller than $10^{-8}$ we make it equal to $+\infty$.

[2]Recall that ARS **V1** and **V2** take in only three hyperparameters: the step-size $\alpha$, the number of perturbation directions $N$, and scale of the perturbations $\nu$. ARS **V1-t** and **V2-t** take in an additional hyperparameter, the number of top directions used $b$ ($b \leq N$).

[3]N/A means that the method did not reach the reward threshold.

[4]UNK stands for unknown.

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
