[Supplementary Material]

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

# A Appendix

## A.1 Reward shaping

For the Hopper-v1, Walker2d-v1, Ant-v1, and Humanoid-v1 tasks the default reward functions include a survival bonus, which rewards RL agents with a constant reward at each timestep, as long as a termination condition (i.e., falling over) has not been reached. For example, the environment Humanoid-v1 awards a reward of 5 at each time step, as long as the Humanoid model does not fall. Hence, if the Humanoid model stands still for 1000 timesteps, it will receive a reward of 5000 minus a small penalty for the actions used to maintain a vertical position. Furthermore, if the Humanoid falls forward at the end of a rollout, it will receive a reward higher than 5000.

It is common practice to report the sample complexity of an RL method by showing the number of episodes required to reach a reward threshold [7, 22, 23]. For example, Gu et al. [7] chose a threshold of 2500, while Rajeswaran et al. [22] chose a threshold of 5280. However, given the survival bonus awarded to Humanoid-v1, we do not believe these reward thresholds are meaningful for locomotion. In Table 1 and Section A.3 we use a reward threshold of 6000 to evaluate the performance of ARS on the Humanoid-v1 task, the threshold also used by Salimans et al. [23].

The survival bonuses awarded by the OpenAI gym discourage the exploration of policies that cause falling early on, which is needed for the discovery of policies that achieve locomotion. These bonuses cause ARS to find policies which make the MuJoCo models stand still for a thousand timesteps; policies which are likely local optima. These bonuses were probably included in the reward functions to help the training of stochastic policies since such policies cause constant movement through stochastic actions. To resolve the local optima problem for training deterministic policies, we subtracted the survival bonus from the rewards outputted by the OpenAI gym during training. For the evaluation of trained policies we used the default reward functions.

## A.2 Supplementary empirical results

**Figure 2:** Showing the standard deviation $\sigma_R$ of the rewards collected at each iteration, while training Humanoid-v1.

**Comparisons to related methods**

Table 3 shows the number of timesteps required by ARS to reach a prescribed reward threshold, averaged over the three fixed random seeds. The hyperparameters were chosen based on the same evaluations performed for Table 1 and Figure 3. We compare ARS to ES and TRPO. For these two methods we show the values reported by Salimans et al. [23], who used six random seeds for evaluation. Salimans et al. [23] do not report sample complexity results for the Ant-v1 and Humanoid-v1 tasks. Table 3 shows that TRPO requires fewer timesteps than ARS to reach the prescribed reward threshold on Walker2d-v1. However, we see that ARS requires fewer timesteps than ES and TRPO on the Swimmer-v1, Hopper-v1, and HalfCheetah-v1 tasks.

Table 4 shows the maximum reward achieved by ARS[5], PPO, A2C, CEM, and TRPO after one million timesteps of the simulator have been collected, averaged over the three fixed random seeds.

**Figure 3:** An evaluation of four versions of ARS on the MuJoCo locomotion tasks. The training curves are averaged over three random seeds, and the shaded region shows the standard deviation. ARS **V2-t** is only shown for the tasks to which it offered an improvement over ARS **V2**.

The hyperparameters were chosen based on the same evaluations performed for Table 1 and Figure 3. Schulman et al. [26] did not report performance of PPO, A2C, CEM, and TRPO on the Ant-v1 and Humanoid-v1 tasks of the OpenAI gym. Table 4 shows that ARS surpasses these four methods on the Swimmer-v1, Hopper-v1, and HalfCheetah-v1 tasks. On the Walker2d-v1 task PPO achieves a higher average maximum reward than ARS, while ARS achieves a similar maximum reward to A2C, CEM, and TRPO.

| | | Average # timesteps to hit Th. | | |
|---|---|---|---|---|
| **Task** | **Threshold** | **ARS** | **ES** | **TRPO** |
| Swimmer-v1 | 128.25 | $6.00 \cdot 10^4$ | $1.39 \cdot 10^6$ | $4.59 \cdot 10^6$ |
| Hopper-v1 | 3403.46 | $2.00 \cdot 10^6$ | $3.16 \cdot 10^7$ | $4.56 \cdot 10^6$ |
| HalfCheetah-v1 | 2385.79 | $5.86 \cdot 10^5$ | $2.88 \cdot 10^6$ | $5.00 \cdot 10^6$ |
| Walker2d-v1 | 3830.03 | $8.14 \cdot 10^6$ | $3.79 \cdot 10^7$ | $4.81 \cdot 10^6$ |

**Table 3:** A comparison of ARS and ES and TRPO methods on the MuJoCo locomotion tasks. For each task we show the average number of timesteps required by ARS to reach a prescribed reward threshold, averaged over three random seeds. For Swimmer-v1 we used ARS **V1**, while for the other tasks we used ARS **V2-t**. The values for ES and TRPO have been averaged over six random seeds and are taken from [23]. Salimans et al. [23] did not evaluate on Ant-v1 and they did not specify the exact number of timesteps required to train Humanoid-v1.

| | | Maximum average reward after # timesteps | | | | |
|---|---|---|---|---|---|---|
| **Task** | **# timesteps** | **ARS** | **PPO** | **A2C** | **CEM** | **TRPO** |
| Swimmer-v1 | $10^6$ | 361 | $\approx 110$ | $\approx 30$ | $\approx 0$ | $\approx 120$ |
| Hopper-v1 | $10^6$ | 3047 | $\approx 2300$ | $\approx 900$ | $\approx 500$ | $\approx 2000$ |
| HalfCheetah-v1 | $10^6$ | 2345 | $\approx 1900$ | $\approx 1000$ | $\approx -400$ | $\approx 0$ |
| Walker2d-v1 | $10^6$ | 894 | $\approx 3500$ | $\approx 900$ | $\approx 800$ | $\approx 1000$ |

**Table 4:** A comparison of ARS, PPO, A2C, CEM, and TRPO on the MuJoCo locomotion tasks. For each task we show the maximum rewards achieved after a prescribed number of simulator timesteps have been used, averaged over three random seeds. The values for PPO, A2C, CEM, and TRPO were approximated based on the figures presented by Schulman et al. [26].

Table 5 shows the maximum reward achieved by ARS, SAC, DDPG, SQL, and TRPO after a prescribed number of simulator timesteps have been collected. The hyperparameters for ARS were chosen based on the same evaluations performed for Table 1 and Figure 3. Table 5 shows that ARS

surpasses SAC, DDPG, SQL, and TRPO on the Hopper-v1 and Walker2d-v1 tasks, and that ARS is surpassed by SAC, DDPG, and SQL on the HalfCheetah-v1 taks. However, ARS performs better than TRPO on this task. On the Ant-v1 task, ARS is surpassed by SAC and performs similarly to SQL, but it outperforms DDPG and TRPO. We did not include values for Swimmer-v1 and Humanoid-v1 because Haarnoja et al. [9] did not use the OpenAI versions of these tasks for evaluation. Instead, they evaluated SAC on the rllab version [5] of these tasks. The authors indicated that Humanoid-v1 is more challenging for SAC than the rllab version because of the parametrization of the states used by the OpenAI gym, and that Swimmer-v1 is more challenging because of the reward function used.

| Task | # timesteps | Maximum average reward after # timesteps | | | | |
|------|-------------|------|------|------|------|------|
| | | **ARS** | **SAC** | **DDPG** | **SQL** | **TRPO** |
| Hopper-v1 | $2.00 \cdot 10^6$ | 3306 | $\approx 3000$ | $\approx 1100$ | $\approx 1500$ | $\approx 1250$ |
| HalfCheetah-v1 | $1.00 \cdot 10^7$ | 5024 | $\approx 11500$ | $\approx 6500$ | $\approx 8000$ | $\approx 1800$ |
| Walker2d-v1 | $5.00 \cdot 10^6$ | 4205 | $\approx 3500$ | $\approx 1600$ | $\approx 2100$ | $\approx 800$ |
| Ant-v1 | $1.00 \cdot 10^7$ | 2072 | $\approx 2500$ | $\approx 200$ | $\approx 2000$ | $\approx 0$ |

**Table 5:** A comparison of ARS, SAC, DDPG, SQL, and TRPO on the MuJoCo locomotion tasks. For each task we show the maximum rewards achieved after a prescribed number of simulator timesteps have been used. The values for ARS were averaged over three random seeds. The values for SAC, DDPG, SQL, and TRPO were approximated based on the figures presented by Haarnoja et al. [9], who evaluated these methods on five random seeds.

## A.3 Computational efficiency

The small computational footprint of linear policies and the embarrassingly parallel structure of ARS make our method ideal for training policies in a small amount of time, with few computational resources. In Tables 6 and 7 we show the wall-clock time required by ARS to reach an average reward of 6000, evaluated over 100 random seeds. ARS requires a median time of 21 minutes to reach the prescribed reward threshold, when trained on one m5.24xlarge EC2 instance with 48 CPUs. The Evolution Strategies method of Salimans et al. [23] took a median time of 10 minutes when evaluated over 7 trials. However, the authors do not clarify what 7 trials means, multiple trials with the same random seed or multiple trials with different random seeds. Moreover, Table 7 shows that ARS trains a policy in at most 10 minutes on 10 out of 100 seeds. Also, Table 6 shows that ARS requires up to 15 times less CPU time than ES.

Finally, we would like to point out that our method could be scaled to more workers. In that case, ARS **V2** will have a computational bottleneck in the aggregation of the statistics $\Sigma_j$ and $\mu_j$ across workers. For successful training of policies, ARS **V2** does not require the update of the statistics (see Line 8 of Algorithm 1) to occur at each iteration. For example, in their implementation of ES, Moritz et al. [18] allowed each worker to have its own independent estimate of $\mu_j$ and $\Sigma_j$. With this choice, the authors used Ray to scale ES to 8192 cores, reaching a 6000 reward on Humanoid-v1 in 3.7 minutes. One could tune an update schedule for the statistics $\Sigma_j$ and $\mu_j$ in order to reduce the communication time between workers or reduce the sample complexity of ARS. For the sake of simplicity we refrained from tuning such a schedule.

| **Algorithm** | **Instance Type** | **# CPUs** | **Median Time** | **CPU Time** |
|---------------|-------------------|------------|-----------------|--------------|
| Evolution Strategies | UNK | 18 | 657 minutes | 197 hours |
| | UNK | 1440 | 10 minutes | 240 hours |
| ARS **V2** | m5.24xlarge | 48 | 21 minutes | 16 hours |
| | c5.9xlarge | 18 | 41 minutes | 12 hours |
| | c4.8xlarge | 18 | 57 minutes | 17 hours |

**Table 6:** An evaluation of the wall-clock time required to reach an average reward of 6000 for the Humanoid-v1 task. The median time required by ARS was evaluated over 100 random seeds. The values for ES were taken from the work by Salimans et al. [23], and were evaluated over 7 independent trials. UNK stands for unknown.

| | | | # minutes by percentile | | | |
|---|---|---|---|---|---|---|
| **Algorithm** | **Instance Type** | **# CPUs** | **10th** | **25th** | **50th** | **75th** |
| | m5.24xlarge | 48 | 10 | 13 | 21 | 45 |
| ARS **V2** | c5.9xlarge | 18 | 16 | 23 | 41 | 96 |
| | c4.8xlarge | 18 | 21 | 28 | 57 | 144 |

**Table 7:** A breakdown by percentile of the number of minutes required by ARS to reach an average reward of 6000 on the Humanoid-v1 task. The percentiles were computed over runs on 100 random seeds.

**Sensitivity to hyperparameters:**

It has been correctly noted in the literature that RL methods should not be sensitive to hyperparameter choices if one hopes to apply them in practice [9]. For example, DDPG is known to be highly sensitive to hyperparameter choices, making it difficult to use in practice [5, 9, 11]. In the evaluations of ARS presented above we used hyperparameters chosen by tuning over the three fixed random seeds. To determine the sensitivity of ARS to the choice of hyperaramers, in Figure 4 we plot the median performance of all the hyperparameters considered for tuning over the three fixed random seeds. Recall that the grids of hyperparameters used for the different MuJoCo tasks are shown in Appendix A.6.

Interestingly, the success rates of ARS depicted in Figure 4 are similar to those shown in Figure 1. Figure 4 shows a decrease in median performance only for Ant-v1 and Humanoid-v1. The similarity between Figures 1 and 4 shows that the success of ARS is as influenced by the choice of hyperparameters as it is by the choice of random seeds. To put it another way, ARS is not highly sensitive to the choice of hyperparameters because its success rate when varying hyperaramers is similar to its success rate when performing independent trials with a "good" choice of hyperparameters. Finally, Figure 4 shows that the performance of ARS on the HalfCheetah-v1 task, a problem often used for evaluations of sensitivity [9, 11], is the least sensitive to the choice of hyperparameter.

**Evaluation of sensitivity to hyperparameters, shown by percentile**

**Figure 4:** An evaluation of the sensitivity of ARS to the choice of hyperparameters. The dotted lines represent median average reward and the shaded regions represent percentiles. We used all the learning curves collected during the hyperparameter tuning performed for the evaluation over the three fixed random seeds. For Swimmer-v1 we used ARS **V1**, and for the rest of the environments we used ARS **V2-t** (and implicitly **V2** when $b = N$).

## A.4 Linear quadratic regulator

While the MuJoCo locomotion tasks considered above are popular benchmarks in the RL literature, they have their shortcomings. The maximal achievable awards are unknown as are the optimal policies, and the current state-of-the-art may indeed be very suboptimal. These methods exhibit high variance making it difficult to distinguish quality of learned policies. And, since it is hard to generate new instances, the community may be overfitting to this small suite of tests.

In this section we propose a simpler benchmark which obviates many of these shortcomings: the classical Linear Quadratic Regulator (LQR) with unknown dynamics. In control theory the LQR with known dynamics is a fundamental problem, which is thoroughly understood. In this problem the goal is to control a linear dynamical system while minimizing a quadratic cost. The problem is formalized in Eq. (3). The states $x_t$ lie in $\mathbb{R}^n$, the actions $u_t$ lie in $\mathbb{R}^p$, and the matrices $A$, $B$, $Q$, and $R$ are have the appropriate dimensions. The noise process $w_t$ is i.i.d. Gaussian. When the dynamics $(A, B)$ are known, under mild conditions, problem (3) admits an optimal policy of the form $u_t = Kx_t$ for some unique matrix $K$, computed efficiently from the solution of an algebraic Riccati equation. Moreover, the finite horizon version of problem 3 can be efficiently solved via dynamic programming.

$$\min_{u_0, u_1, \ldots} \lim_{T \to \infty} \frac{1}{T} \mathbb{E} \left[ \sum_{t=0}^{T-1} x_t^\top Q x_t + u_t^\top R u_t \right] \tag{3}$$
$$\text{s.t.} \ \ x_{t+1} = A x_t + B u_t + w_t$$

LQR with unknown dynamics is considerably less well understood and offers a fertile ground for new research. Note that it is still trivial to produce a varied set of instances for LQR, and we can always compare the best achievable cost when the dynamics are known.

A natural model-based approach consists of estimating the transition matrices $(A, B)$ from data and then solving for $K$ by plugging the estimates in the Riccati equation. A controller $K$ computed in this fashion is called a *nominal controller*. Though this method may not be ideally robust (see, e.g., Dean et al. [4]), nominal control provide a useful baseline to which we can compare other methods.

Consider the LQR instance introduced by Dean et al. [4] as a challenging low-dimensional instance for LQR with unknown dynamics.

$$A = \begin{bmatrix} 1.01 & 0.01 & 0 \\ 0.01 & 1.01 & 0.01 \\ 0 & 0.01 & 1.01 \end{bmatrix}, \ \ B = I, \ \ Q = 10^{-3}I, \ \ R = I. \tag{4}$$

The matrix $A$ has eigenvalues greater than 1, and hence the system is unstable without some control. Moreover, if a method fails to recognize that the system is unstable, it may not yield a stable controller. In Figure 5 we compare ARS to nominal control and to a method using Q-functions fitted by temporal differencing (LSPI), analyzed by Tu and Recht [30].

While Figure 5b shows ARS to require significantly more samples than LSPI to find a stabilizing controller, we note that LSPI requires an initial controller $K_0$ which stabilizes a discounted version of problem (3). ARS does not require a special initialization. However, Figure 5b also shows that the nominal control method is orders of magnitude more sample efficient than both LSPI and ARS. Hence there is much room for improvement for pure model-free approaches.

We conjecture that the LQR instance (4) would also be particularly challenging for policy gradient methods or other methods that explore in the action space. When the control signal is zero, the linear system described by Eq. (4) has a small spectral radius ($\rho \approx 1.024$) and as a result the states $x_t$ would blow up, but slowly. Therefore long trajectories are required for evaluating the performance of a controller. However, the variance of policy gradient methods grows with the length of the trajectories used, even when standard variance reduction techniques are used.

## A.5 Maximum reward achieved after a prescribed number of timesteps

We explain our procedure for obtaining the maximum reward achieved by ARS after a prescribed number of timesteps, averaged over three random seeds. A natural method consists of finding the maximum reward achieved by ARS on each random seed and averaging those values. However, in

**(a)** A comparison of how frequently the controllers produced by ARS, the nominal synthesis procedure, and the LSPI method find stabilizing controllers. The frequencies are estimated from 100 trials. **(b)** A comparison of the relative cost of the controllers produced by ARS, the nominal synthesis procedure, and the LSPI method. The points along the dashed line denote the median cost, and the shaded region covers the 2-nd to 98-th percentile out of 100 trials.

**Figure 5:** A comparison of four methods when applied to the LQR problem (4).

Tables 4 and 5 we compare the performance of ARS to results taken from the figures presented by Schulman et al. [26] and Haarnoja et al. [9], who average training curves across random seeds. For a fair comparison we cannot compare the average of maxima with the maximum of an average of training curves. Therefore, we use the following method for estimating the average maximum reward achieved by ARS.

We begin by introducing some notation. Let $R_i^{(j)}$ be the reward achieved by ARS at iteration $i$, on the $j$th random seed. Also let $h_i^{(j)}$ be the total number of timesteps sampled by ARS up to iteration $i$, on the $j$th random seed. Then, we average the training curves of ARS across the three random seeds to obtain

$$\overline{R}_i = \frac{R_i^{(1)} + R_i^{(2)} + R_i^{(3)}}{3}.$$

If $\beta$ is the prescribed budget of timesteps, let $\overline{h} = \min\{i\,|\,\max\{h_i^{(1)}, h_i^{(1)}, h_i^{(1)}\} \geq \beta\}$. Then, in Tables 4 and 5 we report

$$\overline{R}_{\max} = \max_{0 \leq i \leq \overline{h}} \overline{R}_i. \tag{5}$$

The estimate (5) is a conservative measure of the performance of ARS because $\overline{h}$ is the minimum over the random seeds of the number of iterations needed by ARS to deplete the available budget of timesteps.

## A.6 Hyperparameters

| | Swimmer-v1 | | Hopper-v1 | | HalfCheetah-v1 |
|---|---|---|---|---|---|
| $\alpha$ : | 0.01, 0.02, 0.025 | $\alpha$ : | 0.01, 0.02, 0.025 | $\alpha$ : | 0.01, 0.02, 0.025 |
| $\nu$ : | 0.03, 0.02, 0.01 | $\nu$ : | 0.03, 0.025, 0.02, 0.01 | $\nu$ : | 0.025, 0.02, 0.01 |
| $N$ : | 1 | $N$ : | 8, 16, 32 | $N$ : | 4, 8, 16, 32 |
| $b$ : | 1 | $b$ : | 4, 8, 32 | $b$ : | 2, 4, 8, 32 |
| | Walker-v1 | | Ant-v1 | | Humanoid-v1 |
| $\alpha$ : | 0.01, 0.02, 0.025, 0.03 | $\alpha$ : | 0.01, 0.015, 0.02, 0.025 | $\alpha$ : | 0.01, 0.02, 0.025 |
| $\nu$ : | 0.025, 0.02, 0.01, 0.0075 | $\nu$ : | 0.025, 0.02, 0.01 | $\nu$ : | 0.01, 0.0075 |
| $N$ : | 40, 60, 80, 100 | $N$ : | 20, 40, 60, 80 | $N$ : | 90, 230, 270, 310, 350 |
| $b$ : | 15, 30, 100 | $b$ : | 15, 20, 40, 80 | $b$ : | 100, 200, 360 |

**Table 8:** Grids of hyperparameters used during hyperarameter tuning.

| | **V2** | | | | | **V2-t** | | | | |
|---|---|---|---|---|---|---|---|---|---|---|
| **Task** | $\alpha$ | $\nu$ | $N$ | | **Task** | $\alpha$ | $\nu$ | $N$ | $b$ |
| Swimmer-v1 | 0.02 | 0.01 | 1 | | Swimmer-v1 | 0.02 | 0.01 | 1 | 1 |
| Hopper-v1 | 0.02 | 0.02 | 4 | | Hopper-v1 | 0.01 | 0.025 | 8 | 4 |
| HalfCheetah-v1 | 0.02 | 0.03 | 8 | | HalfCheetah-v1 | 0.02 | 0.03 | 32 | 4 |
| Walker2d-v1 | 0.025 | 0.01 | 60 | | Walker2d-v1 | 0.03 | 0.025 | 40 | 30 |
| Ant-v1 | 0.01 | 0.025 | 40 | | Ant-v1 | 0.015 | 0.025 | 60 | 20 |
| Humanoid-v1 | 0.02 | 0.0075 | 230 | | Humanoid-v1 | 0.02 | 0.0075 | 230 | 230 |

**Table 9:** Hyperparameters for ARS **V2** and **V2-t** used for the results shown Figure 3.