[Reviews · NeurIPS 2018]

Reviewer 1



# Paper ID 914 Simple static linear policies are competitive for RL ## Summary The paper performs a thorough empirical evaluation of the performance of basic randomized search over the class of linear policies on the MuJoCo locomotion tasks. The main idea is to demonstrate the effectiveness of these simple algorithms compared to the much more complex state-of-the-art RL algorithms proposed and evaluated on MuJoCo tasks. The results of the empirical evaluation are startling. The paper convincingly demonstrates very strong performance of the simple algorithm and policy class on the MuJoCo tasks. The evaluation is extremely thorough, the results are compelling and raise serious questions about the current state of RL algorithm evaluation methodology using MuJoCo. In my opinion, this paper is an excellent contribution to the RL literature. ## Detailed Comments - The paper performs a comprehensive evaluation of MuJoCo using a very simple class of linear policies. The learning algorithm is simple randomized search, improved by a few tweaks (reward std. deviation scaling, state normalization, top-k direction filtering). - The main contribution of the paper is to show the strong performance of the above algorithms (ARS) in an extremely thorough evaluation on the MuJoCo control tasks. In most cases, ARS is competitive with the state-of-the-art on the MuJoCo tasks and in some cases (Humanoid-v1, Walker2d-v1), outperforms all known methods. This is a very interesting finding given the sophistication of the RL algorithms, which typically involve the use of deep neural networks with many parameters and hyperparameters, and suggests problems with using MuJoCo as a testbed for evaluating these algorithms. - The 100 seed evaluation is the largest I have seen on the MuJoCo testbed. The variance and local optima, well known to practitioners, is made apparent. This clearly suggests the need for larger seed sets in evaluations for the more sophisticated algorithms but it's not clear to me if the computational overhead of the more involved RL algorithms would even permit such large-scale evaluation. - Overall, the empirical evaluation is extremely thorough and the results point to issues in current RL evaluation methodology. The authors have made the code available and the appendix contains even more experimental detail. I couldn't find much to criticize about this paper. I congratulate the authors on their fine work. Update -------- After reading the reviews and the author responses, I reiterate my original score. As reviewer 3 has pointed out (and the authors have agreed with in their response), having the main contributions clearly stated up front could strengthen the paper.

Reviewer 2



Post rebuttal: I agree with the concerns raised about MujoCo environment used for benchmarking RL algorithms. After reading the other reviews and authors' response I remain unconvinced that they make a good case. My concern is with the methodology of the approach, and I cannot see how it can be addressed in the final version. In the rebuttal, lines 8-12, the authors state: "We offer evidence for the need of new baselines, benchmarks and evaluation methodology for RL algorithms. We show that under current evaluation standards a simple method with linear policies can match or outperform the performance of more complicated methods on current, standard benchmark tasks. Nonetheless, more rigorous evaluation reveals that the assessment of performance on these benchmarks remains overly optimistic because it restricts its attention to too few random seeds." This argument would hold if the paper showed that vanilla random policies reach near RL performance. Instead, the authors tweak and customize algorithms to specific problems. It is the modification of the algorithms that take away from the argument. Hand-coded and highly-custom algorithms generally perform well for specific problems, and the power of more complex learning methods lies in their generality. The authors did not address in rebuttal the differences to Choromanski et al., “Structured Evolution with Compact Architectures for Scalable Policy Optimization,” ICML 2018. This paper uses a more generic and simple algorithm on MojoCo tasks. Methods such in Choromanski et al. can be used to make a better case about the complexity of MujoCo. ------------------ This paper argues for simple baselines on reinforcement learning tasks, and proposes a new such baseline. ARS (Augmented Random Search) is a simple, gradient-free algorithm for training linear policies. The authors compare ARS to a number of RL algorithms, presenting results on sample efficiency, sensitivity, and rewards achieved on MuJoCo locomotion tasks. They redefine “sample efficiency” to include hyperparameter search and argue for more sensitivity analysis in the literature. The paper comes at a good time, with the recent reproducibility issues in RL. The paper’s arguments on sample efficiency and transparency are compelling. The analysis of hyperparameter search and random seed sensitivity sets a good example for the field. The intuition behind the algorithm augmentations is clear. The algorithm, being random search over linear policies, is quite simple and makes me reconsider the need for complex algorithms on these tasks. It is unclear what the main contribution of the paper is: to show that MuJoCo is too simple, to show how RL algorithms should be evaluated, or to introduce a good baseline algorithm? The paper tries to do all of these, but it would be more effective to focus on one and frame the paper around it. For example, if the goal is to introduce a simple, effective baseline, ARS V2-t should be sufficient, without discussing V1. Specifically, in Figure 1, it’s strange to present one algorithm ARS V1 on Swimmer-v1 and use ARS V2 and V2-t on other tasks. It’s somewhat acceptable to use both V2 and V2-t, since as mentioned the difference is just tuning hyperparameter b. In a paper contributing a simple, robust baseline to the field, a single algorithm should work on all tasks? A similar argument could be made against choosing the hyperparameters differently for the sample efficiency and maximum rewards results. In the discussion section, it is argued that “a meaningful sample complexity of an algorithm should inform us on the number of samples required to solve a new, previously unseen task.” Under this definition, it would be nice to see a comparison to metalearning algorithms. Other smaller comments: It would be great if the sensitivity analysis were presented for other algorithms as well, especially on HalfCheetah-v1, as you say this is often used for evaluation of sensitivity. Table 1: bold the most sample efficient algorithm for each task? Table 1: the humanoid-v1 comparison isn’t compelling. Maybe include more results from other algorithms, or remove all discussion of humanoid-v1? In Table 2, it would be great to show the highest reward encountered in the literature for each task, some of what you have in appendix tables 4 and 5. Paper “Structured Evolution with Compact Architectures for Scalable Policy Optimization,” ICML 2018 should be cited as an evidence that more simple, evolutionary policies solve complex tasks. Questions: Why weights initialized to 0, not random? Why grid of hyperparameters instead of random?

Reviewer 3



This paper proposes using random search algorithms for solving RL tasks. They show that with a few modifications of random search, and using linear policy approximators, they achieve competitive results w.r.t policy gradient algorithms and evolution strategy and SOTA in many of them. Arguably, the main contribution of the paper is the results which are interesting. Algorithm-wise, they use a basic random search, which, in turn, augmented by three tricks: i) state normalization, ii) step-length normalization by the standard deviation of returns, and iii) updating based on the highly rewarded population. Definitely, the results of this paper are valuable and I am in favor of these findings, however, I have a few questions/concerns that I would suggest being addressed in the rebuttal/next version. 1) How does a policy learned for one random seed generalize to the other ones? 2) One question is: Does ARS reflect a superior performance in the case of having other policy approximations? For example, is it still working with a NN 1-2 hidden layer? 3) As far as I know, ACKTR surpasses TRPO on Mujuco tasks, so in order to argue SOTA results, ARS should be compared with ACKTR. 4) Although I have not seen a published paper, the following library https://github.com/hardmaru/estool shows that CMA-ES outperforms the ES. One might be interested to see a comparison between ARS and CMA-ES. 5) CMA-ES uses the covariance matrix for choosing the exploration direction in the policy space, while you use it for normalization of the input. Is there any relation between these two? It seems to me that these two ideas are related, using an affine transformation or change of coordinates. Please elaborate. 7) I am confused about the justification of dividing by $\sigma_R$, in general. To illustrate what I mean consider the following example: Let's say at a give iterate, all episode returns are close to each other, then $\sigma_R$ is close to zero. In this case, the update rule 7 in Algorithm 1 will send $M_{j+1}$ to a very far point. Theoretically speaking, instead of $\sigma_R$, $\nu$ should be in the denominator (see Nesterov & Yu, 2011). Minor: footnote 1 in a wrong page. The references to the Appendix inside the text should be B'', not A''. For example, L200. L133: and'' -> an'' How you initialize network weights? ============= After reading the rebuttal, I got the answers to all my question. I have almost no doubt that this paper will have a high impact on the community and will be a motivation for finding harder baselines.